# Design, Synthesis and Biological Evaluation of Pyrazolopyrimidine Derivatives as Aryl Hydrocarbon Receptor Antagonists for Colorectal Cancer Immunotherapy

**DOI:** 10.3390/pharmaceutics17101359

**Published:** 2025-10-21

**Authors:** Byeong Wook Choi, Jae-Eon Lee, Da Bin Jeon, Pyeongkeun Kim, Gwi Bin Lee, Saravanan Parameswaran, Ji Yun Jang, Gopalakrishnan Chandrasekaran, So Yeon Jeong, Geumi Park, Kyoung-jin Min, Heegyum Moon, Jihyeon Yoon, Yerim Heo, Donggun Kim, Se Hwan Ahn, You Jeong Choi, Seong Soon Kim, Jung Yoon Yang, Myung Ae Bae, Yong Hyun Jeon, Seok-Yong Choi, Jin Hee Ahn

**Affiliations:** 1Department of Chemistry, Gwangju Institute of Science and Technology, Gwangju 61005, Republic of Korea; zm1245@gist.ac.kr (B.W.C.); kpk0911@gm.gist.ac.kr (P.K.); hshmhshm@gist.ac.kr (G.B.L.); yjh@gm.gist.ac.kr (J.Y.); yrheo018@gm.gist.ac.kr (Y.H.); lagnaloker0@gm.gist.ac.kr (D.K.); ansehwan@gm.gist.ac.kr (S.H.A.); ujeong2_@gm.gist.ac.kr (Y.J.C.); 2Preclinical Research Center, Daegu-Gyeongbuk Medical Innovation Foundation (K-MEDIhub), Daegu 41061, Republic of Korea; koof12@kmedihub.re.kr (J.-E.L.); jsy111@kmedihub.re.kr (S.Y.J.); pg0929@kmedihub.re.kr (G.P.); jeon9014@kmedihub.re.kr (Y.H.J.); 3Department of Biomedical Sciences, Chonnam National University Medical School, Hwasun 58128, Republic of Korea; dabin4125@gmail.com (D.B.J.); skyjy0505@gmail.com (J.Y.J.); gopalc@gmail.com (G.C.); 4Department of Biotechnology and Bioinformatics, School of Life Sciences, JSS Academy of Higher Education and Research, Mysuru 570015, India; dr.p.saravanan.bi@gmail.com; 5New Drug Development Center, Daegu-Gyeongbuk Medical Innovation Foundation (K-MEDIhub), Daegu 41061, Republic of Korea; kjmin@kmedihub.re.kr (K.-j.M.); heegyum@kmedihub.re.kr (H.M.); 6Therapeutics & Biotechnology Division, Korea Research Institute of Chemical Technology (KRICT), Daejeon 34114, Republic of Korea; kimss@krict.re.kr (S.S.K.); yjy1608@krict.re.kr (J.Y.Y.); mbae@krict.re.kr (M.A.B.); 7JD Bioscience, 208 Cheomdan-dwagiro, Buk-gu, Gwangju 61005, Republic of Korea

**Keywords:** aryl hydrocarbon receptor, antagonist, zebrafish, cancer

## Abstract

**Background:** Aryl hydrocarbon receptor (AhR) is a transcription factor that is involved in the regulation of immunity. AhR inhibits T cell activation in tumors, which induces immune suppression in the blood and solid tumors. We identified effective small-molecule AhR antagonists for cancer immunotherapy. **Methods:** A new series of pyrazolopyrimidine derivatives was synthesized and evaluated for AhR antagonistic activity. **Results:** Compound **7k** exhibited significant antagonistic activity against AhR in a transgenic zebrafish model. In addition, **7k** exhibited good AhR antagonist activity, with a half-maximal inhibitory concentration (IC_50_) of 13.72 nM. Compound **7k** showed a good pharmacokinetic profile with an oral bioavailability of 71.0% and a reasonable half-life of 3.77 h. Compound **7k** selectively exerted anti-proliferative effects on colorectal cancer cells without affecting normal cells, concurrently suppressing the expression of AhR-related genes and the PD-1/PD-L1 signaling pathway. Compound **7k** exhibited potent antitumor activity in syngeneic colorectal cancer models. Importantly, the combination of anti-PD1 and compound **7k** enhanced antitumor immunity by augmenting cytotoxic T lymphocyte (CTL)-mediated activity. **Conclusions:** Collectively, a new pyrazolopyrimidine derivative, **7k**, shows promise as a potential therapeutic agent for treating colorectal cancer.

## 1. Introduction

Cancer is a leading cause of death, with one in five people developing cancer in their lifetime and approximately one in six deaths globally attributed to it. It remains a major obstacle to increasing life expectancy worldwide [1,2]. The immune system’s intricate involvement in cancer development and progression is crucial, given its potential to identify and eliminate abnormal cells, including cancer cells. Immunotherapy is a significant advancement in the treatment of cancer, as it focuses on enhancing the body’s immune system to fight cancer cells [3].

Colorectal cancer is a widespread cancer affecting the colon and rectum. Furthermore, it is the third most frequently diagnosed cancer and the second largest cause of cancer-related deaths worldwide [4]. In colorectal cancer, aryl hydrocarbon receptor (AhR) has recently been shown to play a role in tumorigenesis and has been identified as a potential therapeutic target [5].

AhR is a basic helix–loop–helix (b-HLH)/Per–Arnt–Sim (PAS) transcription factor [6,7]. The receptor consists of the b-HLH domain, which is located at the N-terminus and is related to DNA binding, and the PAS domain [7]. The PAS domain is related to protein dimerization and ligand binding [8]. Without ligand binding, AhR forms a complex with heat shock protein 90 and X-associated protein 2, which remains in the cytoplasm. AhR binds to AhR ligands such as 2,3,7,8-tetrachlorodibenzo-p-dioxin (TCDD; an AhR agonist) in the cytosol, and the AhR-TCDD complex translocates into the nucleus. Subsequently, AhR with a ligand binds to the AhR nuclear translocator and regulates the expression of genes related to the immune system and metabolism [9].

AhR is overexpressed in blood tumors, such as T-cell leukemia and lymphoma, and solid tumors, such as glioblastoma, ovarian cancer, lung cancer, liver cancer, and colorectal cancer [5,10,11,12,13,14,15,16,17,18]. Therefore, AhR can be a prognostic factor for cancer [19]. In addition, AhR is expressed in both cancer cells and tumor-infiltrating immune cells and exerts a direct influence on their responses to the development and progression of cancer, thereby suggesting its involvement in multiple mechanisms that cancer cells utilize for growth and survival [20]. Programmed cell death protein-1 (PD-1) is a protein that acts as an immune checkpoint and plays a crucial role in downregulating the immune system by inhibiting T-cell activation, which allows cancer cells to evade immune detection and destruction [21]. AhR upregulation causes an increase in PD-1 expression in the tumor environment, potentially promoting immune evasion and tumor progression [22] (Figure 1A). Therefore, AhR inhibition can be an effective anticancer strategy.

Diverse AhR antagonists have been reported in the literature and in patents [20,23,24,25,26,27,28,29]. Such antagonists include flavone, azopyrazole, diketone, and fused bicyclic derivatives [20,23,24,25,26,27] (Figure 2). Recently, heterocyclic compounds developed by Bayer, Ikena, and Magenta were reported in patents [20,27,28,29] (Figure 2). In order to identify new AhR antagonists, we chose a series of pyrazolopyrimidine-based compounds, guided by molecular docking for favorable orientation within the receptor’s binding pocket (Figure 1B). Compound **7a** was identified as a hit compound with weak AhR antagonistic activity. In this study, we describe the synthesis and biological evaluation of new pyrazolopyrimidine derivatives as AhR antagonists for cancer immunotherapy.

## 2. Materials and Methods

### 2.1. General Method

All solvents and chemicals were purchased from TCI (Tokyo, Japan), Sigma-Aldrich (St. Louis, MO, USA), Ambeed (Arlington Heights, IL, USA), Combi-Blocks (San Diego, CA, USA), and Duksan Pure Chemicals (Ansan, Republic of Korea), and were used without further purification. All the reported yields are isolated yields after column chromatography or crystallization. ^1^H NMR spectra and ^13^C spectra were recorded on a JEOL JNM-ECS400 spectrometer (JEOL, Tokyo, Japan) at 400 MHz for ^1^H NMR and 100 MHz for ^13^C NMR, respectively. The chemical shift (δ) is expressed in ppm relative to tetramethylsilane (TMS) as an internal standard, and CDCl_3_, MeOH-*d*_4_ and DMSO-*d*_6_ were used as solvents. Multiplicity of peaks is expressed as s (singlet), d (doublet), t (triplet), q (quartet), dd (doublet of doublets), td (triplet of doublets), qd (quartet of doublets), dt (doublet of triplets), and m (multiplet). HRMS data were obtained using a JMS 700 (JEOL, Tokyo, Japan). Optical rotations were measured on a P-2000 polarimeter, purchased from Jasco (Jasco, Tokyo, Japan). High-performance liquid chromatography (HPLC) analyses were performed with a Waters Agilent HPLC system (Agilent Technologies, Santa Clara, CA, USA) equipped with a PDA detector and an Agilent SB-C18 column (1.8 μm, 2.1 × 50 mm). The mobile phase consisted of buffer A (ultrapure H_2_O containing 0.1% trifluoroacetic acid) and buffer B (chromatographic grade CH_3_CN) for method A, and buffer C (chromatographic grade MeOH) for method B was applied at a flow rate of 0.3 mL/min.

*N*-(2-(1*H*-indol-3-yl)ethyl)-5-(3,5-difluorophenyl)-2-isopropyl-2*H*-pyrazolo [4,3-*d*]pyrimidin-7-amine (**7k**)

Step 1. Commercially available 4-nitro-1*H*-pyrazole-3-carboxylic acid (1 g, 6.37 mmol) was dissolved in 100 mL of methanol and 1 mL of sulfuric acid was added. The mixture was heated under reflux for 12 h. After completion of the reaction, the methanol was removed under vacuum. The pH of the solution was adjusted to ~7 by aq.NaOH. Afterwards, the mixture was extracted with water and EA. The combined organic layer was dried over anhydrous Na_2_SO_4_, filtered, and evaporated. The dried extract was concentrated under reduced pressure and purified on silica gel column chromatography to obtain methyl 4-nitro-1*H*-pyrazole-3-carboxylate as a white solid (1.05 g, 96%). ^1^H-NMR (400 MHz, DMSO-*d*_6_) δ 8.39 (s, 1H), 3.80 (s, 3H).

Step 2. Methyl 4-nitro-1*H*-pyrazole-3-carboxylate (3 g, 17.53 mmol) was dissolved in DMF (10 mL). 2-Bromopropane (3.23 g, 26.3 mmol) and potassium carbonate (3.64 g, 26.3 mmol) were added. The reaction mixture was stirred under room temperature for 12 h. After the reaction mixture was cooled to room temperature, the solvent was removed under reduced pressure. The reaction mixture was extracted with ethyl acetate and water. Then the combined organic fraction was dried over anhydrous sodium sulfate and filtered. The solvent was removed under reduced pressure. The residue was then purified via silica gel column chromatography to give methyl 1-isopropyl-4-nitro-1*H*-pyrazole-3-carboxylate (2.09 g, 55%). ^1^H-NMR (400 MHz, DMSO-*d*_6_) δ 9.05 (s, 1H), 4.68–4.58 (m, 1H), 3.88 (s, 3H), 1.45 (d, *J* = 6.7 Hz, 6H).

Step 3. Methyl 1-isopropyl-4-nitro-1*H*-pyrazole-3-carboxylate (2.09 g, 9.78 mmol) was dissolved in MeOH (50 mL). Then, 10% Pd/C (360 mg) was added and the resulting mixture was stirred for 18 h at room temperature under hydrogen atmosphere. The reaction mixture was filtered through celite and the filtrate was concentrated in vacuo and obtained without further purification. Methyl 4-amino-1-isopropyl-1*H*-pyrazole-3-carboxylate (1.37 g, 76%). ^1^H NMR (400 MHz, CDCl_3_) δ 7.25 (s, 1H), 5.28 (s, 2H), 4.15–4.04 (m, 4H), 1.21 (d, *J* = 22.9 Hz, 6H).

Step 4. 4-Amino-1-isopropyl-1*H*-pyrazole-3-carboxylate (1.37 g, 7.48 mmol) was added to urea (2.25 g, 37.39 mmol) and the mixture was heated in a sealed tube at 200 °C for 16 h. The reaction mixture cooled to room temperature and was added to water (50 mL), and the precipitate was filtered to obtain the compound 2-isopropyl-2,4-dihydro-5*H*-pyrazolo [4,3-*d*]pyrimidine-5,7(6*H*)-dione (661 mg, 45%) as an off-white solid which was used in the next step without further purification. ^1^H-NMR (400 MHz, DMSO-*d*_6_) δ 10.82 (s, 2H), 7.68 (s, 1H), 4.62 (td, *J* = 13.4, 6.7 Hz, 1H), 1.43 (d, *J* = 6.4 Hz, 6H).

Step 5. 2-Isopropyl-2,4-dihydro-5*H*-pyrazolo [4,3-*d*]pyrimidine-5,7(6*H*)-dione (120 mg, 0.62 mmol) was added to POCl_3_ (3 mL) under ice-cold conditions, and the resulting mixture was stirred for 12 h at 100 °C. The reaction mixture was cooled to room temperature and concentrated in vacuo and the residue was quenched with ice. The pH of the solution was adjusted to ~7 by aq.NaOH (3–4 mL) and filtered to obtain 5,7-dichloro-2-isopropyl-2*H*-pyrazolo [4,3-*d*]pyrimidine (60 mg, 42%) as a yellow solid without further purification. ^1^H-NMR (400 MHz, DMSO-*d*_6_) δ 9.02 (s, 1H), 5.05–4.95 (m, 1H), 1.59 (d, *J* = 6.5 Hz, 6H).

Step 6. The product obtained above, 5,7-dichloro-2-isopropyl-2*H*-pyrazolo [4,3-*d*]pyrimidine (60 mg, 0.26 mmol), was stirred with tryptamine (62.4 mg, 0.39 mmol) in 25 mL of IPA for 12 h. Afterwards, the mixture was extracted with water and EA. The organic layer was dried with Na_2_SO_4_, and the dried extract was concentrated under reduced pressure. The resulting concentrate was purified on silica gel column chromatography to yield *N*-(2-(1*H*-indol-3-yl)ethyl)-5-chloro-2-isopropyl-2*H*-pyrazolo [4,3-*d*]pyrimidine-7-amine (73 mg, 79%). ^1^H-NMR (400 MHz, DMSO-*d*_6_) δ 9.13–9.27 (1H), 8.56–8.67 (1H), 8.36 (s, 1H), 7.04 (d, *J* = 8.5 Hz, 2H), 6.68 (d, *J* = 8.5 Hz, 2H), 4.77 (s, 1H), 3.61 (d, *J* = 9.2 Hz, 2H), 2.81 (d, *J* = 7.9 Hz, 2H), 1.55–1.52 (m, 6H).

Step 7. *N*-(2-(1*H*-indol-3-yl)ethyl)-5-chloro-2-isopropyl-2*H*-pyrazolo [4,3-*d*]pyrimidine-7-amine (25 mg, 0.07 mmol) and 3,5-difluoro-phenylboronic acid (16.7 mg, 0.11 mmol) were stirred with 2M sodium bicarbonate solution (1 mL) and Pd(PPh_3_)_4_ (8.1 mg, 0.01 mmol) mixture in 10 mL of 1,4-dioxane at 90 °C for 3 h. The reaction mixture was extracted with ethyl acetate and water. The combined organic layer was dried over anhydrous sodium sulfate, concentrated under vacuum. The concentrated product was purified using silica gel column chromatography to yield *N*-(2-(1*H*-indol-3-yl)ethyl)-5-(3,5-difluorophenyl)-2-isopropyl-2*H*-pyrazolo [4,3-*d*]pyrimidin-7-amine (9.3 mg, 31%) as a solid.

^1^H-NMR (400 MHz, MeOD-*d*_4_) δ 8.15 (s, 1H), 7.94–7.89 (m, 2H), 7.68–7.64 (m, 1H), 7.32 (d, *J* = 7.9 Hz, 1H), 7.11–7.06 (m, 2H), 7.02–6.96 (m, 2H), 4.78 (td, *J* = 13.3, 6.6 Hz, 1H), 4.00 (t, *J* = 7.5 Hz, 2H), 3.19 (t, *J* = 7.5 Hz, 2H), 1.60 (dd, *J* = 6.6, 2.6 Hz, 6H); ^13^C-NMR (101 MHz, MeOH-*d*_4_) δ 164.3, 161.8, 156.4, 154.3, 143.3, 138.4, 136.9, 130.3, 127.5, 122.2, 121.1, 121.0, 118.3, 118.1, 112.1, 110.9, 110.5, 110.2, 103.9, 56.0, 41.1, 25.1, 21.9; HRMS (FAB) m/z calculated for C_24_H_22_F_2_N_6_ [M + H]+ 433.1952, found 433.1950; HPLC purity 97.6282%.

### 2.2. Biology

#### 2.2.1. Animals

In vivo experiments were conducted using 6-week-old male immunocompetent BALB/c and C57BL/6 with 18–20 g body weight (Orient Bio Inc., Seungnam, Republic of Korea). All animal experimental procedures were maintained and used in accordance with the Guidelines for the Care and Use of Laboratory Animals of the Institute of Laboratory Animal Center, Daegu-Gyeongbuk Medical Innovation Foundation (K-MEDI hub). The animal studies were conducted after approval by the institutional reviewer board on the Ethics of Animal Experiments of the Daegu-Gyeongbuk Medical Innovation Foundation (approval number: KMEDI-24041103-01).

#### 2.2.2. Zebrafish

Wild-type AB and *Tg(cyp1a:egfp)* zebrafish were obtained from the Korea Zebrafish Resource Center and the Fluorescent Reporter Zebrafish Cooperation Center (no. 1126), respectively, and they were maintained according to the standard procedures and staged in hours post-fertilization (hpf) as per standard criteria [30,31]. Zebrafish embryos were raised at 28.5 °C at all times.

#### 2.2.3. Cells

Chinese hamster ovary CHO cells, mouse fibroblast L929 cells, normal colon CCD18-Co cells, monkey kidney fibroblast COS-7 cells, colorectal cancer cells including HCT-116, RKO, SW480, SW620, CT26, and MC38 cells were purchased from ATCC (American Type Culture Collection). CHO, L929, Cos-7, and all cancer cells were maintained in RPMI 1640 or DMEM supplemented with 10% FBS (Gibco FBS; Thermo Fisher Scientific, Hanover Park, IL, USA) and 1% streptomycin/penicillin. CCD18-Co cells were cultured in high-glucose Dulbecco modified Eagle medium containing 10% FBS and 1% streptomycin/penicillin. All cells were grown in 5% CO_2_ at 37 °C. Mycoplasma testing was regularly performed every month using the BioMycoX Mycoplasma PCR Detection Kit (CellSafe, Yongin, Republic of Korea).

#### 2.2.4. Cell Viability Assay

Cell viability was analyzed using a Cell Counting Kit-8 (Dojindo molecular technologies, Rockville, MD, USA). Cells were seeded in 96-well plates and treated with different concentrations of **7k**. At the indicated time points, CCK-8 (10 μL/well) reagents were added to the cells, followed by further incubation at 37 °C for 90 min. The absorbance at 450 nm was measured using a plate reader (BioTek instruments, Winooski, VT, USA).

#### 2.2.5. qRT-PCR

MC38 and CT26 were treated with different concentrations of **7k** and 200 μM Kynurenine for 24 h. RNA from treated cells was extracted and cDNA was synthesized using the ReverTraAce™ qPCR RT Kit (TOYOBO, Osaka, Japan), followed by evaluation of AhR, AhRR, and IDO-1 mRNA expression with quantitative real-time PCR. PCR conditions were 95 °C preincubation, maintained for 1 min, and then 1 cycle of 95 °C denaturation for 15 s, 55 °C annealing for 15 s, and 72 °C polymerization for 45 s. A total of 40 cycles were repeated. Advanced Relative Quantification analysis was used to confirm the structure of the expected quantitative real-time PCR product, and the variability of other gene amplification was calibrated by relative comparison with GAPDH expression (Table 1).

#### 2.2.6. Cell Nucleus and Cytoplasmic Protein Isolation

Cells were harvested and washed twice in cold PBS. Cytoplasmic and nuclear protein were isolated by NE-PER^TM^ (Thermo Fisher Scientific, Waltham, MA, USA). The procedure was completed according to the kit instructions.

#### 2.2.7. Western Blot

For the extraction of total protein, cells were treated with **7k** for 24 h in a CO_2_ incubator at 37 °C and washed twice with phosphate-buffered saline (PBS). Cell pellets were lysed using radioimmunoprecipitation assay (RIPA) buffer (Thermo Fisher Scientific) containing a protease and phosphatase inhibitor cocktail kit (Thermo Fisher Scientific). The lysed cells were briefly vortexed at intervals and subsequently centrifuged at 13,000× *g* at 4 °C. The protein sample was quantified with the bicinchoninic acid (BCA) protein assay kit (Thermo Fisher Scientific).

Equal amounts of proteins were loaded onto 10% sodium dodecyl sulfate polyacrylamide gel electrophoresis and transferred onto PVDF membranes (Millipore, Billerica, MA, USA). The membrane was blocked with 5% skim milk in Tris-buffered saline (TBS) containing Tween-20 (TBS-T) for an hour and probed with the respective primary antibodies in 5% BSA overnight at 4 °C. Following incubation, the membrane was probed with horseradish peroxidase (HRP)-conjugated secondary antibodies for an hour at room temperature. The membrane was washed three times with TBS-T and the signal was visualized using an enhanced chemiluminescence (ECL) detection reagent (GE Healthcare Life Sciences, Pittsburgh, PA, USA).

The primary antibodies used were as follows: Total p53 (Santa Cruz Biotechnology, Dallas, TX, USA, working dilution 1:500), PARP (CST, Danvers, MA, USA; working dilution 1:1000), cleaved caspase3 (CST, MA, USA; working dilution 1:1000), p21 (CST, MA, USA; working dilution 1:1000), Bcl-2 (CST, MA, USA; working dilution 1:1000), AhR (Santa Cruz; working dilution 1:500), LaminB1 (CST, MA, USA; working dilution 1:2000), GAPDH (Abcam, Cambridge, UK; working dilution 1:5000) and β-actin (CST, MA, USA; dilution 1:5000). HRP-conjugated secondary antibodies used were as follows: anti-mouse (CST, MA, USA) and anti-rabbit (CST, MA, USA).

#### 2.2.8. Cell-Cycle Analysis

Washed and fixed cells were stained in the dark with 0.5 mL propidium iodide/RNase staining buffer for 15 min at room temperature. DNA content, cell-cycle profiles, and forward scatter were analyzed using a Becton Dickinson LSRFortessa^TM^ (BD Diagnostics, San Jose, CA, USA) with emission detection at 488 nm (excitation) and 575 nm (peak emission). Data were analyzed using FlowJo (BD biosciences, San Jose, CA. USA).

#### 2.2.9. Apoptosis Analysis

Cells were washed with PBS containing 1% horse serum and stained with the FITC-Annexin V Apoptosis Detection Kit I (BD Biosciences) at room temperature for 30 min in the dark. DNA content, cell-cycle profiles, and forward scatter profiles were determined using the Becton Dickinson LSRFortessa^TM^ and analyzed with FlowJo software v11.

#### 2.2.10. Clonogenic Assay

Cells were plated into 6-well plates and left for 48 h. After treatment with 5, 10 μM **7k** for 24 h, the drug-containing medium was discarded, and cells were washed twice with PBS and left in complete culture medium for the time corresponding to six doublings. Finally, cells were fixed in 4% Paraformaldehyde (PFA) solution and stained with 0.05% crystal violet.

#### 2.2.11. Flow Cytometry Analysis

For analysis of CD8+ IFNγ + T lymphocytes, splenocytes were collected from tumour-bearing mice and stimulated with a cell activation cocktail (BioLegend, San Diego, CA, USA) in the presence of Brefeldin A for 6 h. Splenocytes were stained with APC-conjugated CD8 and APC-Cy7-conjugated IFNγ Ab at 4 °C for 30 min (BioLegend). After washing with FACS buffer (Biolegend) three times, the cells were analysed using Becton Dickinson LSRFortessa^TM^ and analyzed with FlowJo software. Isotype-matched monoclonal antibodies were used as controls. The data were analysed using Flowjo analysis software v11. (Flowjo LLC, Ashland, OR, USA).

For analysis of PD-L1 expression in colorectal cancer cells, MC38 in 24-well plates were treated with IFNγ (10 ng/mL) for 24 h and incubated with 5 and 10 μM for 24 h. Cells were collected and stained with APC-conjugated PD-L1 Ab at 4 °C for 30 min (BioLegend). After washing with FACS buffer (Biolegend) three times, the cells were analysed using Becton Dickinson LSRFortessa^TM^ and analyzed with FlowJo software. Isotype-matched monoclonal antibodies were used as controls. The data were analysed using Flowjo analysis software (FlowJo LLC, Ashland, OR, USA).

For analysis of PD-1 expression in PD-1-overexpressing Jurkat T cells (Jurkat/PD-1 cells), Jurkat/PD-1 cells in 24-well plates were treated with 2 μg/mL Phytohaemagglutinin (PHA) and incubated with 5 and 10 μM **7k** for 24 h. Cells were collected were stained with APC-conjugated PD-1 Ab at 4 °C for 30 min (BioLegend). After washing with FACS buffer (Biolegend) three times, the cells were analysed using Becton Dickinson LSRFortessa^TM^ and analyzed with FlowJo software. Isotype-matched monoclonal antibodies were used as controls. The data were analysed using Flowjo analysis software (FlowJo LLC, Ashland, OR, USA).

#### 2.2.12. Cytotoxic T Lymphocyte (CTL) Assay

Splenocytes were collected from tumour-bearing mice and stimulated with a cell activation cocktail (BioLegend) for 6 h. MC38/Luc cells as target cells were seeded in black, clear-bottomed 96-well plates (1 × 10^4^ cells). After overnight incubation, splenocytes as effectors were added into wells containing target cells at an effector to target ratio of 10:1. LLC/Luc and CT26/Luc cells were used as negative target cells. Eleven hours later, luciferase activity was measured with an IVIS Lumina III instrument (PerkinElmer, Waltham, MA, USA).

#### 2.2.13. In Vivo Therapy

Study 1: Antitumor activity of **7k** alone in mice with either CT26 syngeneic model or MC38 syngeneic model. Mice were subcutaneously challenged with 5 × 10^5^ either MC38 or CT26 cells. When tumor formation was detected via inspection and palpation, **7k** treatment was initiated once a day for 14 days by oral gavage.

Study 2: Antitumor activity of combination with anti-PD1 and **7k** in CT26 tumor bearing mice. Mice were subcutaneously challenged with 5 × 10^5^ CT26 cells. Tumor bearing mice were divided into four groups as following; Group 1: vehicle, Group 2: 10 mg/kg anti-PD1, Group 3: 10 mg/kg **7k**, and Group 4: 10 mg/kg anti-PD1 + 10 mg/kg **7k**. **7k** was administered by oral gavage once daily. Anti-PD1 antibody (clone RPM1-14, Bio X cell, Lebanon, NH, USA) was injected intraperitoneally every 3 days for 4 doses.

The **7k** was dissolved in solution including 5% DMSO, 55% DW, and 40% PEG. Tumor size was measured with a caliper at the indicated time points, and tumor volume (mm^3^) was calculated using the following formula, tumor volume (mm^3^) = d2 × D/2, where d and D are the shortest and longest diameters in mm, respectively.

#### 2.2.14. Molecular Docking Studies of the Synthesized Compounds with Human AhR

The structure of human AhR PAS-B domain is not determined experimentally in the X-ray structure (PDB Id: 5NJ8). Considering the low resolution of Cryo-EM structure, and the influence of protein binding partners Hsp90 and XAP2 on the structure of human AhR PAS-B domain and the possibility of the binding of small molecules (inhibitors), the cryo-EM structure is not preferred. Therefore, the homology model of human AhR PAS-B domain reported by Ashley J Parks et al. is consistent with the published literature and utilized for molecular docking studies. The structures of synthesized compounds were manually drawn, and their respective 3D conformation was generated after energy minimization with ChemOffice package. The structures of synthesized compounds and AhR model were preprocessed for docking calculations with AutoDockTools 1.5.7. The grid that encompasses the ligand binding pocket of PAS-B domain was constructed for the binding prediction of synthesized compounds with 300 initial conformations. The docking calculations were carried out with rigorous maximum evaluations of 50,000,000 for 200 runs. The binding pose of synthesized compounds was chosen based on the better predicted free energy of binding from the largest cluster. The intermolecular interactions of synthesized compounds with human AhR were investigated with Discovery Studio Visualizer 25.

#### 2.2.15. Pharmacokinetic Study in Mouse

To estimate the pharmacokinetic parameters in animals, the drug was administered via intravenous (I.V., 5 mg/kg) and oral (P.O., 10 mg/kg) routes in seven-week-old male ICR mice (Orient Bio Inc., Seongnam, Korea). Blood samples were collected at 0.083 (for I.V.), 0.5, 1, 2, 4, 8, and 24 h after drug administrations and then immediately centrifuged at 10,000× *g* for 3 min. The plasma concentrations of the drug were determined via LC-MS/MS. The plasma concentration–time profiles and pharmacokinetic parameters were estimated using the non-compartmental method with the nonlinear least squares regression program WinNonlin 5.3 (Pharsight, Mountain View, CA, USA).

#### 2.2.16. AhR Antagonist Screening Using Tg(cyp1a:egfp) Zebrafish

At 30 hpf, *Tg(cyp1a:egfp)* embryos were placed in a 2-mL screw-capped glass vial with five embryos per vial. Each vial contained 1 mL of E3 media (5 mM NaCl, 0.17 mM KCl, 0.33 mM CaCl_2_·2H_2_O and 0.4 mM MgCl_2_·6H_2_O, pH 7.2) and supplemented with 2 nM TCDD (1746-01-6; AccuStandard Inc., New Haven, NH, USA) dissolved in DMSO. The compounds were subsequently diluted and individually introduced into the vials, achieving a final concentration of 0.1% DMSO. The embryos were then incubated for 48 h and then embryonic EGFP expression was evaluated using a fluorescent stereomicroscope (SteREO Discovery V8; Zeiss, Oberkochen, BW, Germany). As a positive control, StemRegenin1 (72342, STEMCELL Technologies, Vancouver, BC, Canada) was employed.

#### 2.2.17. Determination of the Half Maximal Inhibitory Concentration (IC_50_) of Compounds

All experiments were conducted as described previously unless stated otherwise [26]. COS-7 cells were seeded onto 24-well plate until approximately 65% confluency. The cells were transfected with ARNT-1c expression plasmid (50 ng), AhR2 expression plasmid (5 ng), Cyp1a-firefly luciferase reporter plasmid (20 ng) and pRL-TK Renilla luciferase plasmid (2 ng) using Lipofectamine 3000 (Thermo Fisher Scientific) according to the manufacturer’s protocol. The pEGFP-C1 plasmid (Clontech, Tempe, AZ, USA) was used to assess the transfection efficiency. At 5 h post transfection, serially diluted test compounds at indicated concentration were added to the cells. Simultaneously, TCDD (10 nM) or indicated compounds with the highest concentration were added in the control wells. At 1 h post compound treatment, TCDD (10 nM) was added to all wells except the control wells. At 18 h post TCDD treatment, the medium was removed, and the cells washed with Dulbecco’s phosphate buffered saline (DPBS) and harvested using 1x Passive Lysis Buffer (Promega, Madison, WI, USA). The dual luciferase assay was carried out using a Dual-Luciferase Reporter Assay System (Promega, Madison, WI, USA) and an Orion L Microplate Luminometer (Titertek Berthold, Pforzheim, BW, Germany) as per manufacturer’s protocol. The relative light unit (RLU) values were normalized and expressed as toxic induced ratio (TIR). IC_50_ was determined using Prism 9.0 (GraphPad Software).

## 3. Results

### 3.1. Chemistry

A new series of pyrazolopyrimidine derivatives was synthesized, as depicted in Figure 1 and Figure 2. Commercially available 4-nitro-1*H*-pyrazole-3-carboxylic acid underwent esterification, followed by alkylation with bromoalkanes to give **2a**–**c**. Compounds **2a**–**c** were reduced under a hydrogen atmosphere in the presence of Pd/C, followed by cyclization using urea under reflux conditions to give **4a**–**c**. Compounds **4a**–**c** were chlorinated with phosphorus oxychloride to give **5a**–**c**.

Compounds **5a**–**c** were coupled with tryptamine derivatives to give **6a**–**c**, followed by Suzuki coupling with a substituted phenyl boronic acid to afford the final compounds **7a**–**m**.

### 3.2. SAR Data of AhR Antagonists

To evaluate AhR antagonistic activity, we conducted an AhR antagonist assay using transgenic zebrafish [26,32]. Zebrafish have garnered significant attention as model organisms for drug discovery because they share metabolic pathways with mammals. In our previous study, we successfully generated *Tg(cyp1a:egfp)* zebrafish, in which enhanced green fluorescent protein (EGFP) expression exhibited dose-dependent patterns across various tissues such as the gut, blood vessels, cloaca, liver, pronephros, pectoral fin bud, pancreas, and swimming bladder upon exposure to the potent AhR ligand, TCDD [26,32]. The antagonistic activity was measured using EGFP as a visual readout, and StemRegenin 1 (SR-1) was used as the reference standard [33]. Compound **7a**, which was a hit, showed AhR antagonistic activity with + (based on fluorescence in *Tg (cyp1a:egfp)* zebrafish). First, we optimized phenyl moiety at 5-position of pyrazolopyrimidine and the results are summarized in Table 2. p-Cyanophenyl derivative **7b** exhibited better activity compared with those of the hit compound (**7a**). m-Cyanophenyl derivative **7c** showed improved activity (++++) with an IC_50_ value of 171 nM, which was more potent than that of SR-1 (reference standard, +++, IC_50_ value of 198 nM). Therefore, we synthesized more meta substituted phenyl derivatives. However, m-acetyl (**7d**), m-trifluoromethyl (**7e**), m-amino (**7f**) didn’t show AhR inhibition (Table 2). We also synthesized heteroaromatic derivatives including pyrimidine and pyridine at 5-position; however, their activities were decreased.

Based on the results shown in Table 1, we optimized pyrazolopyrimidine with m-cyanophenyl groups by introducing several alkyl chains to the pyrazole ring. Ethyl (**7g**) and n-butyl (**7h**) derivatives showed AhR inhibition and had lower activity than that of the isopropyl compound (**7b**). When tryptamine was modified to serotonin (**7i**), compound **7i** showed decreased AhR inhibition in the zebrafish model (Table 3).

As a continuation of the modification, we further optimized the *m*-cyanophenyl moiety, retaining the tryptamine and isopropyl moieties, and the results are summarized in Table 3. *m*-Fluoro derivative (**7j**) showed potential (++++) with an IC_50_ value of 412 nM. Moreover, 3,5-difluorophenyl derivative dramatically increased AhR inhibition (+++++) with an IC_50_ value of 13.72 nM. Other derivatives (**7l** and **7m**) showed decreased activities; therefore, we chose **7k** for further evaluation (Table 4, Figure 3).

### 3.3. PK Data of 7k

Next, we conducted an in vivo pharmacokinetic (PK) study, and the results are summarized in Table 4. Compound **7k** exhibited good oral area under the curve of 8.16 ± 3.67 µg·h/mL and a reasonable half-life (T_1/2_) of 3.77 h. The bioavailability of compound **7k** was estimated to be 71% after oral administration (PO) (Table 5).

### 3.4. Docking Study

The docking studies revealed that binding occurred in the ligand-binding pocket of the PAS-B domain of human AhR as observed with a known antagonist SR-1 but with a different intermolecular interaction. The estimated free energy of binding (docking score) of the synthesized compound **7k** and StemRegenin-1 are −10.87 kcal/mol and −11.92 kcal/mol, respectively. StemRegenin-1 showed slightly better docking scores than the synthesized compound **7k** as the hydroxyl group of StemRegenin-1 formed additional hydrogen bonds with Tyr332 and His337 of human AhR PAS-B domain (Appendix A). StemRegenin-1 has relatively more hydrophilic interactions, especially hydrogen bonds, while the synthesized compound **7k** had predominantly hydrophobic interactions. This is usually observed as docking scoring functions give relatively less weightage to hydrophobic interactions than hydrophilic interactions while estimating the free energy of binding. The synthesized compounds predominantly exhibited hydrophobic intermolecular interactions with AhR, which contributed significantly to the stabilization of the compounds in the ligand-binding pocket. The five-membered ring of tryptamine moiety in compound **7k** formed a π–Donor Hydrogen bond with the hydroxyl group of Thr289 of human AhR (Figure 4). Furthermore, the tryptamine moiety in compound **7k** had π–π interactions with Tyr310 and Phe324, and hydrophobic interactions with Pro297, Cys300, Leu308, Leu315, and Leu353 (Figure 4 and Appendix A). The isopropyl moiety in the pyrazole ring of compound **7k** exhibited hydrophobic interactions with Phe287, Phe351, Leu353, and Val363 (Figure 4 and Appendix A). Moreover, Phe351 had π–π interactions with the pyrazolopyrimidine moiety, which interacted with Val381 via hydrophobic interactions and Met348 via π–sulfur bonds (Figure 4 and Appendix A). Collectively, the hydrophobic interactions of the tryptamine, isopropyl, pyrazolopyrimidine and phenyl moieties of compound **7k** optimized the binding mode with human AhR in an extended conformation, which was not observed with compounds **7e**, **7f** and **7i**. The different binding modes of the compounds **7e**, **7f** and **7i** could be attributed to the steric hinderance by functional groups attached to the phenyl group (Appendix A).

The differences in the intermolecular interactions of the functional groups in the phenyl moiety of the synthesized compounds showed significant differences in the structure–activity relationship of the synthesized compounds with AhR inhibition. Halogen bonds were observed between Cys333 and Ser336 with fluorine at the fifth position of the 3,5-difluorophenyl moiety of compound **7k**; this was not observed for the 3-fluorophenyl moiety of compound **7j**. The phenyl ring of compound **7k** had hydrophobic interactions with Met340, Val381, Ala367, and Phe395, along with π–sigma bonds with Met348 and π–sulfur bonds with Cys333 of human AhR (Figure 4 and Appendix A). It is also notable that the positioning of functional groups attached to phenyl groups of the compounds **7c**, **7g** and **7h** are not optimally placed to make interactions with Cys333 and Ser336 of human AhR. This leads to a slight deviation of the pyrazolopyrimidine and tryptamine moiety of the compounds **7c** and **7h** from the binding pose of compound **7k**, which leads to significant changes in their intermolecular interactions with human AhR, while compounds **7g** and **7j** share similar binding pose with compound **7k** (Appendix A). Altogether, the halogen and π–sulfur bonds with hydrophobic interactions formed by the 3,5-difluorophenyl moiety improved the inhibition of AhR by compound **7k**.

### 3.5. Antiproliferative Effects in Colorectal Cancer Cells and Not in Normal Cells

To determine the anticancer effects of **7k**, we examined its effects on the proliferation of normal and colorectal cancer cells. After incubation with **7k** for 24 h, cell viability was determined using the cell counting kit 8 assay. As shown in Figure 5B, **7k** induced a dose-dependent decrease in cell proliferation in colorectal cancer cells. However, **7k** exhibited few antiproliferative effects in normal cells (Figure 5A). Similarly, SR-1, a known AHR antagonist, reduced the viability of colorectal cancer cells but had little effect on normal colon CCD-18Co cells (Appendix A). Consistently, the clonogenic assay demonstrated a significant decrease in the colony formation ability of colorectal cancer cells treated with **7k** compared with that in vehicle-treated cells (Figure 5C). These results indicate that **7k** induces selective cytotoxicity in colorectal cancer cells.

### 3.6. 7k-Induced Cell Cycle Arrest and Apoptosis Activation in Colorectal Cancer Cells

Considering that we observed **7k**-mediated anti-proliferation effects in colorectal cancer cells, such as CT26 and MC38 cells, we further investigated the status of the cell cycle and apoptosis in **7k**-treated cancer cells using flow cytometry. Cell cycle analysis revealed a dose-dependent increase in the sub-G1 fraction in **7k**-treated colorectal cancer cells and not in vehicle-treated colorectal cancer cells (Figure 6A,B and Appendix A). Fluorescence-activated cell sorting analysis with Annexin V and propidium iodide demonstrated an increase in apoptosis in **7k**-treated colorectal cancer cells in a dose-dependent manner (Figure 6C,D and Appendix A).

We conducted western blotting to determine the status of **7k**-induced apoptosis-regulating proteins in colorectal cancer cells, including MC38 and CT26 cells. As depicted in Figure 7, treatment with **7k** resulted in the upregulation of p53, a tumor suppressor, in colorectal cancer cells [34,35]. Consistent with the cell cycle arrest findings, an increase in p21, a cell cycle inhibitor, was observed in **7k**-treated colorectal cancer cells. Furthermore, the downregulation of Bcl-2, an anti-apoptotic gene, was observed in **7k**-treated colorectal cancer cells. Treatment with **7k** resulted in the upregulation of cleaved poly (ADP-ribose) polymerase and cleaved caspase 3, which are important effectors of apoptosis in colorectal cancer cells. These findings suggest that **7k** induces cell cycle arrest and apoptosis in colorectal cancer cells by activating the p53/p21/caspase-3 signaling pathway.

### 3.7. Inhibition of Kynurenine-Induced AhR Downstream Gene Expression and Its Nuclear Translocation by 7k

To determine whether the anticancer effects of **7k** were mediated through AhR inhibition, we evaluated the expression of AhR downstream targets, including AhR, AhRR, and IDO-1, in kynurenine-stimulated cancer cells using qPCR analysis. Our results showed that kynurenine, an AhR agonist, significantly upregulated the expression of AhR and its downstream targets, including AhRR and IDO-1, in MC38 and CT26 cells (Figure 8A,B). In contrast, treatment with **7k** reduced the expression levels of AhR, AhRR, and IDO-1 in kynurenine-stimulated cancer cells in a dose-dependent manner.

To further examine the effects of **7k** on kynurenine-induced AhR nuclear translocation, Western blot analysis was conducted. As shown in Figure 9, treatment with kynurenine led to a reduction in AhR protein levels in the cytoplasm, while an increase in its expression was observed in the nucleus. In contrast, the addition of **7k** restored AhR protein levels in the cytoplasm and decreased AhR expression in the nucleus. The finding indicated that **7k** effectively downregulated the expression of AhR downstream targets and inhibited the nuclear translocation of the AhR protein.

### 3.8. 7k-Mediated Suppression of PD-1/PD-L1 Protein Expression

The PD-1/PD-L1 pathway is a critical regulator of immune tolerance within the tumor microenvironment. Several studies have demonstrated that AhR modulators can influence PD-1/PD-L1 signaling in the tumor microenvironment, thereby enhancing the efficacy of immune checkpoint therapy. Based on this, we investigated whether **7k** could modulate PD-1/PD-L1 signaling in both cancer cells and T lymphocytes. First, we assessed the effects of **7k** on PD-L1 expression in cancer cells using FACS analysis. Consistent with previous reports, treatment with IFNγ significantly induced PD-L1 expression in MC38 and CT26 cancer cells (Figure 10A–D). Interestingly, **7k** caused a dose-dependent downregulation of PD-L1 expression in both cancer cell lines. Next, we examined the effects of **7k** on PD-1 expression in lymphocytes using PD-1-overexpressing Jurkat T cells (Jurkat/PD-1 cells). Phytohaemagglutinin (PHA), a selective T-cell mitogen, is known to enhance PD-1 expression in Jurkat T cells. As shown in Figure 10E,F, FACS analysis using an anti-PD1 antibody confirmed high PD-1 expression in Jurkat/PD-1 cells. As expected, treatment with PHA further increased PD-1 expression in Jurkat/PD-1 cells compared to vehicle-treated controls. Notably, **7k** reduced PD-1 expression in PHA-treated Jurkat/PD-1 cells in a dose-dependent manner. These findings demonstrate that **7k** uniquely inhibits PD-1/PD-L1 expression in both cancer cells and immune cells.

### 3.9. Antitumor Activity of 7k in Immunocompetent Mice with Colorectal Cancer

We examined the antitumor activity of **7k** as a single agent in immunocompetent mice of either MC38 or CT26 syngeneic model. As depicted in Figure 11A, tumor-bearing mice orally received **7k** once a day for 14 days, followed by the measurement of tumor volumes. As a results, oral administration of **7k** led to noticeable inhibition of tumor growth, showing 33% and 49% of tumor growth inhibition (TGI) in CT26 and MC38 tumor-bearing mice, respectively (Figure 11B–E). Significant differences in tumor volume were first noticeable between **7k**-treated mice and vehicle-treated mice at day 10 post-treatment. Consistently, tumor weight was significantly lower in **7k**-treated mice compared to vehicle-treated mice (Appendix A). We further evaluated the anticancer efficacy of **7k** in comparison to the known AhR antagonist 3′,4′-dimethoxyflavone using a syngeneic MC38 model. As shown in Figure 11F,G, both 3′,4′-dimethoxyflavone and **7k** significantly inhibited tumor growth. Notably, the antitumor effect was more pronounced in **7k**-treated mice compared to 3′,4′-dimethoxyflavone-treated mice, with tumor growth inhibition (TGI) rates of 37% and 51% in the 3′,4′-dimethoxyflavone and **7k** treatment groups, respectively. The results indicate the possibility of using **7k** as a therapeutic agent against colorectal cancer in vivo.

### 3.10. Antitumor Activity of Combination with 7k and Immune Checkpoint Inhibitor Anti-PD1 in Immunocompetent Mice with Colorectal Cancer

Many studies have demonstrated that AhR is an essential target to overcome immunosuppression, and the modulation of AhR using its inhibitors leads to enhanced antitumor immunity in vivo, thereby promoting the therapeutic efficacy of immune checkpoint inhibitors [22,36,37,38]. Based on the intriguing finding that **7k** downregulates PD-1/PD-L1 signaling and exhibits significant antitumor effects, we further explored its potential in combination therapy with the immune checkpoint inhibitor and the anti-PD1 antibody in immunocompetent mice with either CT26 and MC38 syngeneic model. The in vivo therapy was performed as shown in Figure 12A. As a result, in vivo administration of either anti-PD1 or **7k** alone led to a retardation of tumor growth. A significant difference in tumor volume was found in **7k**-treated mice but not in anti-PD1-tretaed mice compared with vehicle-treated mice (Figure 12B,C). The combination of anti-PD1 and **7k** induced significantly greater antitumor effects than those in either of the single agents alone, which was consistent with the tumor weight measurements (Appendix A). The TGI was 38%, 49%, and 68% in the anti-PD1 group, **7k** group, and anti-PD1+**7k** group, respectively. As depicted in Figure 13A, we further investigated the antitumor effects of the combination of **7k** and anti-PD1 therapy in another syngeneic colorectal cancer model, the MC38 syngeneic model. Consistently with the findings from the CT26 syngeneic model, monotherapy with either anti-PD1 or **7k** showed mild antitumor activity, with significant differences compared to the vehicle-treated group (Figure 13B,C and Appendix A). Notably, the combination with anti-PD1 and **7k** exhibited enhanced antitumor activity in the MC38 tumor model. The TGI for MC38 tumor model was 39%, 45%, and 64% in the anti-PD1 group, **7k** group, and anti-PD1+**7k** group, respectively. No abnormal reduction in body weight was observed in **7k**-treated mice during in vivo therapy (Appendix A). These findings support the potential use of **7k** as an enhancer of anti-PD1 immune checkpoint inhibitors in vivo.

We finally examined the antitumor immunity generated by the combination of **7k** and anti-PD1 therapy. The increased levels of cytotoxic T cell (CTLs), essential immune effectors in eliciting antitumor immune responses, plays a critical role in the successful induction of antitumor effects in vivo. Effector CD8+T cells produce IFN-γ and granzyme, which are pivotal in inducing apoptosis in target cancer cells. Therefore, we investigated IFNγ-producing CD8 T lymphocytes using flow cytometry analysis. FACS analysis revealed that monotherapy with either anti-PD1 or **7k** increased the levels of CD8+ IFNγ T lymphocytes compared to vehicle-treated mice (Figure 14A,B). Notably, the combination therapy significantly enhanced the levels of CD8+ IFN-γ-producing T lymphocytes compared to either single therapy.

We next explored the killing activity of cytotoxic T cells (CTLs) derived from splenocytes of tumor-bearing mice using luciferase-expressing cancer cells (MC38-luc cells). The luciferase-based bioluminescent imaging system enabled us to easily monitor CTL-mediated killing activity by measuring the luciferase activity in target cancer cells. Unlike the FACS analysis results showing the number of CD8+ IFNγ T cells, splenocytes from mice treated with monotherapy (either anti-PD1 or **7k**) exhibited minimal killing activity against MC38-luc cells (Figure 14C,D). In contrast, splenocytes from the combination therapy group significantly reduced luciferase activity in MC38-luc cells, indicating robust CTL-mediated cytotoxic activity. These findings underscore the potential of **7k** as an enhancer of anti-PD1 immune checkpoint inhibitors in vivo by promoting CTL-mediated antitumor immunity.

## 4. Discussion

The aryl hydrocarbon receptor (AhR) plays a pivotal role in modulating immune responses and tumor progression, and recent studies have positioned AhR as a compelling target for immunotherapy. In this study, we designed and synthesized a series of pyrazolopyrimidine derivatives and identified **7k** as a potent AhR antagonist with promising anticancer properties.

The structure–activity relationship (SAR) analysis demonstrated that meta-substitution on the phenyl ring significantly affected AhR antagonistic activity. Compound **7c**, bearing a meta-cyano group, showed improved efficacy compared to the hit compound **7a** and even outperformed the reference compound SR-1. However, the introduction of other meta-substituents, including acetyl, trifluoromethyl, and amino groups, led to a complete loss of activity. These findings suggest that electronic and steric contributions at the meta-position are critical for optimal binding affinity and receptor inhibition. Further optimization of this scaffold led to the identification of **7k**, the 3,5-difluorophenyl derivative, which exhibited the most potent AhR inhibition (IC_50_ = 13.72 nM).

Molecular docking analysis provided further mechanistic insight into the superior AhR antagonistic activity of compound **7k**. Docking simulations using a validated homology model of the human AhR PAS-B domain revealed that **7k** forms multiple stabilizing interactions, including halogen bonding between the difluorophenyl group and polar residues within the ligand-binding pocket, as well as π–sulfur and hydrophobic interactions with key residues. These interactions were predicted to contribute to the enhanced binding affinity of **7k** compared to less active analogs such as **7g** and **7j**, which lacked the same level of docking stability and interaction complexity. Notably, the predicted binding pose of **7k** was consistent with its potent biological activity observed in zebrafish and cellular assays, suggesting that the structural features identified through SAR and docking cooperatively drive receptor antagonism. This convergence of in silico and in vitro findings reinforces the significance of the difluorophenyl moiety and supports the utility of rational design in optimizing AhR-targeted agents.

Beyond receptor binding, **7k** exerted potent biological effects on colorectal cancer cells. The compound selectively suppressed proliferation in various colorectal cancer cell lines while sparing normal cells, demonstrating high tumor selectivity. Cell cycle analysis and apoptosis assays revealed that **7k** induced sub-G1 arrest and increased apoptotic populations. Mechanistically, **7k** upregulated phosphorylated p53 and p21 while downregulating anti-apoptotic Bcl-2, thereby activating the intrinsic apoptotic pathway through caspase-3 cleavage. These findings align with the notion that **7k** exerts direct cytotoxic effects in cancer cells via the p53/p21/caspase-3 axis.

Importantly, **7k** reversed the kynurenine-induced upregulation of AhR downstream genes, including AhR, AhRR, and IDO-1, and inhibited AhR nuclear translocation. These results confirm that **7k** functions as a true AhR antagonist, restoring immune surveillance pathways typically suppressed in the tumor microenvironment. Furthermore, **7k** modulated immune checkpoint signaling by downregulating PD-L1 expression in cancer cells and suppressing PD-1 expression in T cells, a dual immunomodulatory effect that expands its therapeutic relevance.

Pharmacokinetic analysis demonstrated that **7k** possesses drug-like characteristics, including an oral bioavailability of 71% and a half-life of 3.77 h. These features suggest that **7k** can achieve sufficient systemic exposure via oral administration, making it suitable for long-term therapeutic use.

In syngeneic mouse models of colorectal cancer, **7k** significantly suppressed tumor growth in both CT26 and MC38 models. When combined with anti-PD1 antibody, the therapeutic efficacy was markedly enhanced compared to either treatment alone. In the MC38 model, the combination therapy achieved tumor growth inhibition of up to 68%, exceeding the effects of **7k** (49%) or anti-PD1 (38%) monotherapy. Similarly, in the CT26 model, the combination treatment led to superior tumor control relative to single-agent treatments. These consistent results across two distinct immunocompetent models underscore the robust potential of **7k** with immune checkpoint blockade. Furthermore, combination therapy increased IFNγ-producing CD8+ T cell populations and enhanced CTL-mediated cytotoxicity, suggesting that **7k** augments antitumor immunity through both T cell activation and direct tumor targeting.

In summary, the compound **7k** represents a novel, orally available AhR antagonist that exerts potent anticancer activity through both direct cytotoxic effects and immune system modulation. Its dual mechanism of inhibiting AhR and PD-1/PD-L1 pathways, combined with favorable pharmacokinetics and strong in vivo efficacy, highlights its promise as a therapeutic candidate for colorectal cancer, particularly in combination with existing immunotherapies. Further studies on safety and efficacy in humanized models are warranted.

## 5. Conclusions

In this study, new pyrazolopyrimidine derivatives were developed for cancer treatment. Among the derivatives, compound **7k** showed good AhR antagonist activity, with an IC_50_ value of 13.72 nM. Moreover, compound **7k** had a noteworthy in vivo PK profile with 71% bioavailability after PO administration with a T_1/2_ of 3.77 h. Compound **7k** effectively modulated the AhR and AhR downstream genes, thereby down-regulation of PD-L1/PD-1 signaling in cancer cells and CD8 T cells. Furthermore, compound **7k** exhibited selective cytotoxicity against colorectal cancer cells and induced apoptosis via p53/p21/caspase 3 signaling, thereby leading to in vivo antitumor activity when orally administered as a single agent. The combination of compound **7k** and anti-PD1 exhibited enhanced antitumor effects in immunocompetent mice with CT26 tumors via generation of strong antitumor immunity of CTLs. In conclusion, a new pyrazolopyrimidine derivative **7k** shows promise as a potential therapeutic agent against colorectal cancer, either as a single treatment or in combination with immune checkpoint inhibitors.

## Data Availability

The datasets used and/or analyzed during the current study are available from the corresponding author upon request.

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
