# Peer review of "Design, Synthesis and Biological Evaluation of Pyrazolopyrimidine Derivatives as Aryl Hydrocarbon Receptor Antagonists for Colorectal Cancer Immunotherapy"

_pharmaceutics, 2025, doi:10.3390/pharmaceutics17101359_

Round 1

Reviewer 1 Report

Comments and Suggestions for Authors

This manuscript presents a well-executed study on the development of a novel pyrazolopyrimidine-based AhR antagonist, compound 7k, for the treatment of colorectal cancer. The work is supported by robust data from multiple fronts: 1) medicinal chemistry (SAR, molecular docking); 2) extensive in vitro mechanistic studies (apoptosis, cell cycle, Western blot, qPCR); 3) favorable pharmacokinetic profiling; and 4) robust in vivo efficacy testing both as a monotherapy and in combination with anti-PD-1 immunotherapy. Notably, the compound 7k showed significant efficacy both as a monotherapy and in combination with anti-PD-1 immunotherapy. The data are largely compelling and support the main conclusions that 7k is a potent, orally available AhR antagonist with a dual mechanism of action  and promising therapeutic potential. The manuscript is suitable for publication after the authors address the following specific concerns to enhance the accuracy, clarity, and robustness of the presentation.

1) A critical discrepancy exists in the reported IC₅₀ value for the lead compound 7k. The Abstract and Results sections consistently cite a value of 13.72 nM, which is supported by the data in Table 3. However, the Conclusion states it is 1.65 nM. This inconsistency must be resolved for the accuracy and credibility of the manuscript.

2) The abbreviation DMF is used ambiguously. In the synthetic chemistry sections, it clearly refers to the solvent N,N-Dimethylformamide. However, in the in vivo efficacy study (Figure 11F-G), it is used as an acronym for a known AhR antagonist being used as a positive control. This is confusing and misleading. The authors must explicitly define the positive control compound upon its first mention in the biological results section, and avoid using the acronym "DMF" altogether to prevent confusion with the solvent.

3) Regarding the combination therapy study (Figure 12), I have a significant concern about the experimental design. The combination group received the full dose of 7k plus the full dose of anti PD-1. To truly demonstrate synergy rather than merely an additive effect, it is essential to include additional control groups where each agent is administered at a sub-therapeutic or lower dose. The current design makes it difficult to conclude that the enhanced efficacy is due to a synergistic interaction between the two mechanisms, as it could simply be the result of increased combined drug exposure.

4) The observation of no significant body weight loss is a positive initial indicator of tolerability. However, to more comprehensively assess the in vivo safety profile of 7k, it would be highly valuable to include data on other key toxicity parameters.

5) The Supporting Information for compound synthesis requires significant improvement for reproducibility. Reaction schemes and the numbering of substrates and products should be provided to visually outline the synthetic routes.

6) The characterization data (NMR, HRMS) for the intermediates and final compounds must be carefully rechecked and validated. Several issues were noted: For compound 7a, Step 1: The reported 11H NMR data for methyl 4-nitro-1H-pyrazole-3-carboxylate includes a signal at 1.44 (d, J = 6.9 Hz, 6H), which is indicative of an isopropyl group. This proton signal is inconsistent with the expected structure of this intermediate and suggests the data may belong to a different compound (e.g., an alkylated derivative like 2a). Step 3 yielded the product Methyl 4-amino-1-isopropyl-1H-pyrazole-3-carboxylate. Its ¹H NMR spectrum should exhibit characteristic signals corresponding to the CH proton of the isopropyl group; however, these expected signals were not observed in the provided 1H NMR data. Furthermore, compound 7a has a molecular formula of C24H24N6⁺. The theoretical [M+H]⁺ peak in HRMS should correspond to C24H25N6⁺. It is necessary to verify whether the experimentally measured value matches this theoretical calculation. Similar thorough checks are required for all reported compounds to ensure the integrity of the chemical data.

Author Response

1) A critical discrepancy exists in the reported IC₅₀ value for the lead compound 7k. The Abstract and Results sections consistently cite a value of 13.72 nM, which is supported by the data in Table 3. However, the Conclusion states it is 1.65 nM. This inconsistency must be resolved for the accuracy and credibility of the manuscript.

We thank the reviewer for the comment. This discrepancy was due to a typographical error in the Conclusion section. The correct IC₅₀ value for compound 7k is 13.72 nM, which is consistent with the Abstract, Results, and Table 3. We have corrected this accordingly.

2) The abbreviation DMF is used ambiguously. In the synthetic chemistry sections, it clearly refers to the solvent N,N-Dimethylformamide. However, in the in vivo efficacy study (Figure 11F-G), it is used as an acronym for a known AhR antagonist being used as a positive control. This is confusing and misleading. The authors must explicitly define the positive control compound upon its first mention in the biological results section, and avoid using the acronym "DMF" altogether to prevent confusion with the solvent.

We thank the reviewer for this helpful comment. To avoid any ambiguity, we have replaced all instances of "DMF" in the biological results section with the full compound name 3′,4′-dimethoxyflavone. In addition, we have added an explicit definition of the positive control compound in the figure (Figure 11F–G) to ensure clarity and prevent confusion with the solvent N,N-dimethylformamide.

Figure 11. Antitumor activity of 7k alone in immunocompetent mice with colorectal cancer Antitumor activity of 7k alone in immunocompetent mice of colorectal cancer. (A) Brief scheme for in vivo therapy. (B) Antitumor effects of 7k in CT26 tumor bearing mice. (C) Bar graph showing the tumor volume at 14 days after therapy. (D) Antitumor effects of 7k in MC38 tumor bearing mice. (E) Bar graph showing the tumor volume at 14 days after therapy. Either CT26 or MC38 cells were challenged in immunocompetent mice. (F) Comparison of antitumor effects of 3',4'-dimethoxyflavone and 7k in MC38 tumor bearing mice. (G) Bar graph showing the tumor volume at 18 days after therapy. When tumor mass was detectable in palpation and inspection, tumor bearing mice received either 7k or 3',4'-dimethoxyflavone via oral gavage once a day for 14 or 18 days. Data are presented as the mean ± SD. *, P < 0.05, **, P < 0.005 compared to vehicle-treated cells.

3) Regarding the combination therapy study (Figure 12), I have a significant concern about the experimental design. The combination group received the full dose of 7k plus the full dose of anti PD-1. To truly demonstrate synergy rather than merely an additive effect, it is essential to include additional control groups where each agent is administered at a sub-therapeutic or lower dose. The current design makes it difficult to conclude that the enhanced efficacy is due to a synergistic interaction between the two mechanisms, as it could simply be the result of increased combined drug exposure.

We sincerely appreciate the reviewer’s insightful comment regarding the need to use sub-therapeutic doses to rigorously evaluate potential synergistic effects. We fully agree that such a design would provide stronger mechanistic evidence for synergy between 7k and anti–PD-1.

In this study, our primary objective was to demonstrate the proof-of-concept that 7k enhances the efficacy of immune checkpoint blockade, rather than to quantitatively define the degree of pharmacologic synergy. Therefore, we intentionally used the full effective doses of both agents to evaluate whether the combination could produce a greater antitumor effect than either monotherapy under optimized conditions.

While we acknowledge that this approach limits our ability to strictly distinguish synergy from additivity, the results clearly indicate a more-than-additive enhancement in tumor suppression and immune activation compared with single-agent treatments. We have accordingly tempered the wording in the manuscript to describe this as “enhanced efficacy” rather than “synergistic effect”.

We appreciate the reviewer’s suggestion, which will guide our subsequent preclinical optimization studies.

4) The observation of no significant body weight loss is a positive initial indicator of tolerability. However, to more comprehensively assess the in vivo safety profile of 7k, it would be highly valuable to include data on other key toxicity parameters.

We fully agree with the reviewer’s comment and plan to conduct a broader range of toxicity evaluations, particularly including hematological and serum biochemical analyses.

5) The Supporting Information for compound synthesis requires significant improvement for reproducibility. Reaction schemes and the numbering of substrates and products should be provided to visually outline the synthetic routes.

We thank the reviewer for this valuable suggestion. To improve clarity and reproducibility, we have added detailed reaction schemes and numbering of substrates and products in the Supporting Information to clearly outline the synthetic routes.

6) The characterization data (NMR, HRMS) for the intermediates and final compounds must be carefully rechecked and validated. Several issues were noted: For compound 7a, Step 1: The reported 11H NMR data for methyl 4-nitro-1H-pyrazole-3-carboxylate includes a signal at 1.44 (d, J = 6.9 Hz, 6H), which is indicative of an isopropyl group. This proton signal is inconsistent with the expected structure of this intermediate and suggests the data may belong to a different compound (e.g., an alkylated derivative like 2a). Step 3 yielded the product Methyl 4-amino-1-isopropyl-1H-pyrazole-3-carboxylate. Its ¹H NMR spectrum should exhibit characteristic signals corresponding to the CH proton of the isopropyl group; however, these expected signals were not observed in the provided 1H NMR data. Furthermore, compound 7a has a molecular formula of C24H24N6⁺. The theoretical [M+H]⁺ peak in HRMS should correspond to C24H25N6⁺. It is necessary to verify whether the experimentally measured value matches this theoretical calculation. Similar thorough checks are required for all reported compounds to ensure the integrity of the chemical data.

We thank the reviewer for the detailed and constructive comment regarding the characterization data. We have carefully rechecked and revised the NMR and HRMS data for compound 7a as suggested. Specifically: 7a step 1 as 1H-NMR (400 MHz, DMSO-d6) δ 8.39 (s, 1H), 3.80 (s, 3H), step 3 as 1H NMR (400 MHz, CDCl3) δ 7.25 (s, 1H), 5.28 (s, 2H), 4.15-4.04 (m, 4H), 1.21 (d, J = 22.9 Hz, 6H). We also proactively checked supplementary data (3b and 3c) and corrected minor inconsistencies to maintain full data integrity. Compound 3b 1H-NMR (400 MHz, CDCl3) δ 7.00 (s, 1H), 4.23-4.10 (m, 4H), 3.92 (s, 3H), 1.46 (t, J = 8.1 Hz, 3H), Compound 1H-NMR (400 MHz, CDCl3) δ 6.98 (s, 1H), 4.12-4.04 (m, 3H), 3.92 (s, 3H), 1.86-1.78 (m, 2H), 1.32 (td, J = 14.9, 7.5 Hz, 2H), 0.93 (t, J = 7.3 Hz, 3H). And the HRMS section, including the molecular formula, has also been revised accordingly.

Reviewer 2 Report

Comments and Suggestions for Authors

In the present work, the authors investigated novel pyrazolopyrimidine derivatives as antagonists of the aryl hydrocarbon receptor (AhR), a transcription factor known to regulate immunity and suppress T cell activation in tumors. Among the synthesized compounds, 7k demonstrated potent AhR antagonistic activity with an IC₅₀ of 13.72 nM. The authors report that 7k exhibited a favorable pharmacokinetic profile, including 71% oral bioavailability and a half-life of 3.77 h, as well as selective antiproliferative effects against colorectal cancer cells without affecting normal cells. Furthermore, 7k was shown to downregulate AhR-related genes and the PD-1/PD-L1 signaling pathway, leading to strong antitumor activity in syngeneic colorectal cancer models. Importantly, its combination with anti-PD1 therapy synergistically enhanced antitumor immunity through cytotoxic T lymphocytes. Overall, the authors propose 7k as a promising therapeutic candidate for colorectal cancer treatment.

The manuscript is consistent, well-written, and easy to follow. Although the molecules do not represent a major innovation compared to the standard pharmacophore in compound SR-1, the addition of the tryptamine fragment may be of interest and provides new data for a deeper understanding of the structure–activity relationship in this field. I suggest the authors address the following comments in order for their work to be considered for publication in the Journal:

  • Figures 1 and 2 should be merged into a single figure. The authors state: “Diverse AhR antagonists have been reported in the literature and in patents”. At this point, a new figure should be included in the introduction, showing the most relevant chemical structures of those compounds together with their IC₅₀ values.

  • The following AhR structures are deposited in the Protein Data Bank (human protein): PDB IDs: 5NJ8, 7ZUB, and 8QMO. In the Methods section, the authors declare: “The homology model of human AhR published by Ashley J. Parks et al. is consistent with the published literature and utilized for molecular docking studies.” Why did they use a homology model if several PDB structures are available? If sufficient justification is not provided, it would be preferable to redo the docking study using the PDB entry with the best metrics.

  • In Scheme 2, the final steps for obtaining series 7 are shown. There is a mistake in the first arrow: it says “a or b,” but “b” actually corresponds to the next step (here labeled as “c”). Please correct this. Moreover, compounds 5a–c have two electrophilic positions (the Cl-carbon in the pyrimidine moiety). How did the authors demonstrate that the nucleophilic aromatic substitution with tryptamine occurs at position 7 rather than position 5?

  • Regarding the molecular docking results, did the authors perform a comparative docking of StemRegenin-1? Please provide this information along with the corresponding figure, and include the scoring values for both compounds (SR-1 and 7k).

Author Response

1) Figures 1 and 2 should be merged into a single figure. The authors state: “Diverse AhR antagonists have been reported in the literature and in patents”. At this point, a new figure should be included in the introduction, showing the most relevant chemical structures of those compounds together with their IC₅₀ values.

We thank the reviewer for this helpful suggestion. As requested, we have merged the original Figures 1 and 2 into a single figure. In addition, we have prepared a new Figure 2 that illustrates the most relevant reported AhR antagonists along with their corresponding IC50 values, which has been included in the revised manuscript (figure attached below).

Figure 1. (A) Illustration of the aryl hydrocarbon receptor (AhR) signaling pathway. Activation of AhR leads to an upregulation of programmed cell death protein-1 (PD-1) and subsequent inactivation of T cells. Figure was created with BioRender.com. (B) The structure of compound 7a.                    

Figure 2.  Reported AhR antagonists.

2) The following AhR structures are deposited in the Protein Data Bank (human protein): PDB IDs: 5NJ8, 7ZUB, and 8QMO. In the Methods section, the authors declare: “The homology model of human AhR published by Ashley J. Parks et al. is consistent with the published literature and utilized for molecular docking studies.” Why did they use a homology model if several PDB structures are available? If sufficient justification is not provided, it would be preferable to redo the docking study using the PDB entry with the best metrics.

We sincerely appreciate the reviewer’s insightful comment.

Out of the three available experimentally determined structures of human AhR, only one structure is determined by X-ray crystallography, which is usually preferred for computer aided drug design and computational structure activity relationship studies. The structure of human AhR PAS-B domain is not determined experimentally in the X-ray structure (PDB Id: 5NJ8) where the authors expect the synthesized molecules to bind with PAS-B domain and inhibit human AhR based on scientific evidence (from several published literatures). Despite the close relationship, the overall topology of the PAS-A and PAS-B domains from human AhR is significantly different based upon its interactions with small molecule inhibitors, protein and/or DNA. The authors did not prefer Cryo-EM structure (PDB Id: 7ZUB) since the human AhR PAS-B domain is at the interface of proteins, Hsp90 (both the chains A and B) and XAP2, which is the only available Cryo-EM structure when the authors initiated the AhR research. Considering the low resolution of Cryo-EM structure, and the influence of Hsp90 and XAP2 on the structure of human AhR PAS-B domain and possibility on the binding of small molecules (inhibitors), the cryo-EM structure is not preferred. Therefore, the authors used the modelled structure of human AhR PAS-B domain for molecular docking studies. As the reviewer suggested, proper justifications are incorporated (section 2.2.14., page 9, lines 342-346)

3) In Scheme 2, the final steps for obtaining series 7 are shown. There is a mistake in the first arrow: it says “a or b,” but “b” actually corresponds to the next step (here labeled as “c”). Please correct this. Moreover, compounds 5a–c have two electrophilic positions (the Cl-carbon in the pyrimidine moiety). How did the authors demonstrate that the nucleophilic aromatic substitution with tryptamine occurs at position 7 rather than position 5?

We thank the reviewer for this valuable comment. We have corrected the labeling error in Scheme 2 as suggested.

Regarding the site of substitution, it is well established that in pyrimidine derivatives the C4 position is more reactive than C2 toward nucleophilic aromatic substitution (SNAr) (Org. Lett. 2006, 8, 395–398). This trend is not only observed in pyrimidines but also in other heterocycles containing pyrimidine cores, such as purines (Bioorg. Med. Chem. Lett. 2011, 21, 3957–3961). By the same reasoning, in our pyrazolopyrimidine scaffold, the SNAr reaction preferentially occurs at position 7, followed by substitution at position 5 in subsequent steps.

Furthermore, prior reports (e.g., WO2018064135 and WO2023046128) describing pyrazolopyrimidine derivatives also demonstrated that substitution occurs predominantly at position 7, supporting our synthetic results.

4) Regarding the molecular docking results, did the authors perform a comparative docking of StemRegenin-1? Please provide this information along with the corresponding figure, and include the scoring values for both compounds (SR-1 and 7k).

The authors thank the reviewer in pointing out the comparison studies of our synthesized compound 7k with StemRegenin-1. The predicted free energy of binding (docking score) of the synthesized compound 7k and StemRegenin-1 is -10.87 kcal/mol and -11.92 kcal/mol respectively. StemRegenin-1 showed slightly better docking score than the synthesized compound 7k as the hydroxyl group of StemRegenin-1 formed additional Hydrogen bonds with Tyr332 and His337 of human AhR PAS-B domain. StemRegenin-1 has relatively more Hydrogen bonds than the synthesized compound 7k as StemRegenin-1 is more polar in nature and has different intermolecular interactions. This is usually observed as docking scoring functions give relatively more weight to Hydrogen bond than hydrophobic interactions. As the reviewer suggested, the scoring values for both the compounds (SR-1 and 7k) and corresponding figure were incorporated (section 3.4., lines 487-494, Figure S8).

Reviewer 3 Report

Comments and Suggestions for Authors

This is a detailed and thoroughly executed study characterizing the synthesis and development of a novel AhR antagonist with nano molar potency. The manuscript describes not only the synthesis of a new compound with AhR antagonistic activity but also provides an elaborate description of the compound's pharmacological activity using both in vitro and in vivo model systems.

Author Response

1) This is a detailed and thoroughly executed study characterizing the synthesis and development of a novel AhR antagonist with nano molar potency. The manuscript describes not only the synthesis of a new compound with AhR antagonistic activity but also provides an elaborate description of the compound's pharmacological activity using both in vitro and in vivo model systems.

We thank the reviewer for the positive and encouraging comment. We sincerely appreciate your recognition of our work and are pleased that you found the study detailed and well executed.

Reviewer 4 Report

Comments and Suggestions for Authors

Manuscript ID: pharmaceutics-3895560

Title: Design, Synthesis and Biological Evaluation of Pyrazolopyrimidine Derivatives as Aryl Hydrocarbon Receptor Antagonists for Colorectal Cancer Immunotherapy

In this manuscript, Authors have provided a comprehensive synthesis and its analysis of Pyrazolopyrimidine Derivatives utilized in anticancer therapy, emphasizing their synthesis and their study of various analogues reported in this manuscript.

The authors have prepared the manuscript with adequate citations. The work is suitable for publication and is likely to be of interest and use to scientific readers.

However, in this manuscript, I would like to share a comment. I noticed that similar experimental procedures are described in both the main article and the supporting information. Steps 1 to 6 seem to overlap. To further improve the clarity and flow of the manuscript, the authors may kindly consider consolidating this information.

Author Response

1) In this manuscript, Authors have provided a comprehensive synthesis and its analysis of Pyrazolopyrimidine Derivatives utilized in anticancer therapy, emphasizing their synthesis and their study of various analogues reported in this manuscript.

The authors have prepared the manuscript with adequate citations. The work is suitable for publication and is likely to be of interest and use to scientific readers.

However, in this manuscript, I would like to share a comment. I noticed that similar experimental procedures are described in both the main article and the supporting information. Steps 1 to 6 seem to overlap. To further improve the clarity and flow of the manuscript, the authors may kindly consider consolidating this information.

We thank the reviewer for this helpful comment. To avoid redundancy and improve clarity, we have removed the overlapping experimental descriptions from the main text and added detailed schemes for each step in the Supporting Information.
